# 3-Hydroxytanshinone Inhibits the Activity of Hypoxia-Inducible Factor 1-α by Interfering with the Function of α-Enolase in the Glycolytic Pathway

**DOI:** 10.3390/molecules29102218

**Published:** 2024-05-09

**Authors:** Tae Hyun Son, Shin-Hye Kim, Hye-Lim Shin, Dongsoo Kim, Hwan Gyu Kim, Yongseok Choi, Sik-Won Choi

**Affiliations:** 1School of Life Sciences and Biotechnology, Korea University, Seoul 02841, Republic of Korea; snoopyegg@korea.ac.kr; 2Forest Biomaterials Research Center, National Institute of Forest Science (NIFoS), Jinju 52817, Republic of Korea; black7a@korea.kr (S.-H.K.); hlims0901@korea.kr (H.-L.S.); skimds@korea.kr (D.K.); 3Department of Biological Sciences, Jeonbuk National University, Jeonju 54896, Republic of Korea; hgkim@jbnu.ac.kr

**Keywords:** 3-Hydroxytanshinone, hypoxia, HIF-1α, α-enolase, glycolysis

## Abstract

Tumor cells in hypoxic conditions control cancer metabolism and angiogenesis by expressing HIF-1α. Tanshinone is a traditional Chinese medicine that has been shown to possess antitumor properties and exerts a therapeutic impact on angiogenesis. However, the precise molecular mechanism responsible for the antitumor activity of 3-Hydroxytanshinone (3-HT), a type of tanshinone, has not been fully understood. Therefore, our study aimed to investigate the mechanism by which 3-HT regulates the expression of HIF-1α. Our findings demonstrate that 3-HT inhibits HIF-1α activity and expression under hypoxic conditions. Additionally, 3-HT inhibits hypoxia-induced angiogenesis by suppressing the expression of VEGF. Moreover, 3-HT was found to directly bind to α-enolase, an enzyme associated with glycolysis, resulting in the suppression of its activity. This inhibition of α-enolase activity by 3-HT leads to the blockade of the glycolytic pathway and a decrease in glycolysis products, ultimately altering HIF1-α expression. Furthermore, 3-HT negatively regulates the expression of HIF-1α by altering the phosphorylation of AMP-activated protein kinase (AMPK). Our study’s findings elucidate the mechanism by which 3-HT regulates HIF-1α through the inhibition of the glycolytic enzyme α-enolase and the phosphorylation of AMPK. These results suggest that 3-HT holds promise as a potential therapeutic agent for hypoxia-related angiogenesis and tumorigenesis.

## 1. Introduction

In 2022, the number of new cancer cases and deaths was estimated that there were 20 million new cases of cancer and 9.7 million cancer-related deaths. It is estimated that approximately 20% of individuals will develop cancer during their lifetimes, resulting in mortality rates of one in nine for men and one in 12 for women [1]. The resolution of cancer treatment remains incomplete. Tumors encounter a range of stressors, including hypoxia and nutrient deprivation [2]. To evade these stressors, tumor cells utilize diverse mechanisms to ensure their survival. In the presence of hypoxia, tumor cells employ the Hypoxia Inducible Factor-1α (HIF-1α) to control the expression of multiple genes related to cell survival, including glycolysis, in order to generate and regulate vital elements necessary for cell survival [3]. HIF-1α also governs the regulation of vascular endothelial growth factor (VEGF) to stimulate angiogenesis, a process that facilitates the delivery of oxygen and nutrients to the cells [4]. Glycolysis, a metabolic process controlled by HIF-1α, encompasses a series of enzymatic reactions and serves as the principal mechanism through which cancer cells produce adenosine triphosphate (ATP), the essential energy currency of cells, enabling their survival [5]. Thus, the regulation of HIF-1α could serve as a potential approach for treating cancer cells.

Glycolysis is a metabolic pathway that entails the catalytic action of several enzymes to degrade glucose into pyruvate, yielding 2 ATP molecules as a byproduct [6]. Cancer cells are known to induce the activation of HIF-1α under hypoxic conditions, leading to the upregulation of glycolysis-associated proteins like GLUT-1, thereby facilitating the sustenance of the glycolytic pathway. This phenomenon is referred to as the Warburg effect [7]. α-enolase, a pivotal enzyme in glycolysis, plays a critical role in enhancing cell proliferation and motility in cancer cells. The overexpression of this gene specifically results in heightened cell migration and enhanced resistance to drugs in cancer cells [8]. Glycolysis plays a crucial role in the regulation of ATP levels, enabling the cellular energy sensor AMPK to modulate energy levels [9]. Elevated ATP concentrations resulting from glycolysis impede the activation of AMP-activated protein kinase (AMPK), whereas decreased ATP levels due to glycolysis suppression stimulate the activation of AMPK. Activated AMPK has the capacity to modulate energy levels and act as an inhibitor of energy metabolism in tumor cells [10]. Hence, glycolysis and its related pathways can be a focal point for cancer therapy.

Natural products exhibit inherent stability and offer a rich source of inspiration for novel drug discovery, due to their diverse chemical structures [11]. Furthermore, there is a growing interest in the functional food industry in regard to the stability and effectiveness of these products [12]. Tanshinone belongs to the group of abietane diterpenes, initially discovered by Nakao in the 1930s in the roots of the traditional Chinese medicinal herb known as ‘tanshen’. More than 40 derivatives of tanshinone, such as 3-Hydroxytanshinone (3-HT), have been identified through isolation processes [13]. Tanshinone has exhibited diverse therapeutic effects, including cardiovascular, neurological, anti-cancer, and osteoporosis preventive properties [14,15,16,17]. Nevertheless, the anticancer properties of 3-HT remain poorly understood, and the molecular mechanisms underlying the anticancer effects of tanshinone are not well elucidated. Hence, a research study was carried out to clarify the impact of 3-HT on cancer and to explore the mechanisms that drive its anti-cancer properties.

## 2. Results

### 2.1. 3-HT Suppresses the Hypoxia-Induced Expression of HIF-1α

Tanshinone derivatives were evaluated through the use of a HIF-1α luciferase assay (Appendix A). Among these derivatives, 3-HT (Figure 1A) exhibited the most significant inhibition of luciferase activity under conditions that mimic hypoxia induced by cobalt chloride (CoCl_2_) (Figure 1B) and did not display any cytotoxic effects at the specified concentrations (Figure 1C). 3-HT demonstrated a concentration-dependent inhibition of HIF-1α, Glut-1, and VEGF mRNA expressions (Figure 1D). Furthermore, 3-HT showed a concentration-dependent reduction in CoCl_2_-induced HIF-1α protein expression. The expression of HIF-1α protein in a 1% oxygen environment was also suppressed by 3-HT (Figure 1E). In summary, 3-HT suppressed the activity and expression of HIF-1α under hypoxic conditions in HeLa cells.

### 2.2. 3-HT Attenuates Hypoxia-Induced Angiogenesis in Both In Vitro and Ex Vivo

To examine the influence of 3-HT on hypoxia-induced angiogenesis, we conducted an in vitro capillary assay using HUVECs induced by CoCl_2_. The results indicated that 3-HT significantly inhibited the formation of tubes in a dose-dependent manner (Figure 2A). The concentration of 3-HT did not affect its cytotoxicity, as demonstrated in Figure 2B. Additionally, the transcriptional level of VEGF was reduced by 3-HT (Figure 2C). To further investigate the anti-angiogenic effects of 3-HT, an ex vivo chick embryo chorioallantoic membrane (CAM) experiment was performed. The experiment showed that the 3-HT concentration dependently inhibited new blood vessel growth in chorioallantoic membranes without any observed toxicity or adverse effects (Figure 2D). The inhibition of neovessel formation by 3-HT was concentration-dependent, as indicated by the proportion of chick embryos demonstrating anti-angiogenesis (Figure 2E). The findings indicate that both the in vitro and ex vivo studies yielded evidence supporting the inhibitory effect of 3-HT on angiogenesis.

### 2.3. 3-HT Interacts with the α-Enolase Protein, Leading to the Inhibition of Its Enzymatic Activity

To identify the target molecules of 3-HT, biotinylated 3-HT was immobilized on a streptavidin column and then exposed to total proteins extracted from HeLa cells. The target protein was subsequently isolated through SDS-PAGE gel electrophoresis using elution with 3-HT. The target protein was isolated from the gel, enzymatically cleaved with trypsin, and subjected to sequencing through liquid chromatography-tandem mass spectrometry (LC-MS/MS). The peptide fragment demonstrated homology with α-enolase (Figure 3A). To confirm the specificity of the binding of 3-HT to α-enolase, purified α-enolase was spotted onto nitrocellulose membranes. α-enolase was found to be elevated at the indicated concentrations (Figure 3B, top panel). The utilization of biotin led to a dose-dependent increase in biotinylated 3-HT, which consequently interacts with α-enolase. When 3-HT and biotinylated 3-HT were co-treated, no detection of either compound was evident in the biotin immunoblots (Figure 3B). To assess the binding kinetics and affinity between α-enolase and 3-HT, a surface plasmon resonance (SPR) assay was conducted. Following the conjugation of biotinylated 3-HT to NLC chips, enolase was employed as the analyte and flowed over the surface at different concentrations. In chips coated with biotinylated 3-HT, the refractive index exhibited a concentration-dependent increase with the concentration of α-enolase, as shown in Figure 3C. To examine the impact of the binding between α-enolase and 3-HT on the catalytic activity of α-enolase, an enzymatic activity assay was performed. 3-HT suppressed the activity of α-enolase in a dose-dependent fashion (Figure 3D). These results suggest that 3-HT directly binds to α-enolase, leading to the inhibition of its activity.

### 2.4. 3-HT Inhibits the Expression of HIF-1α by Directly Modulating the Activity of α-Enolase

To demonstrate the interaction between α-enolase and HIF-1α, loss-of-function studies were performed to evaluate the expression of HIF-1α in correlation with the expression of α-enolase. The siRNA targeting α-enolase resulted in a decrease in HIF-1α mRNA expression under normoxic conditions and conditions mimicking hypoxia induced by cobalt chloride (CoCl_2_) (Figure 4A). Moreover, siRNA against α-enolase exhibited attenuation of HIF-1α protein expression in conditions simulating hypoxia induced by cobalt chloride (CoCl_2_) (Figure 4B). In the investigation of the glycolysis pathway rescue experiment, Krab’s buffer supplemented with either glucose or pyruvate was utilized.

Upon exposure to glucose instead of pyruvate, the suppression of HIF-1α expression was reduced by siRNA targeting α-enolase under both normoxic and hypoxia-mimicking conditions (Figure 4C). The mRNA expression of HIF-1α was reduced by 3-HT in a concentration-dependent manner following glucose treatment. However, pyruvate treatment did not demonstrate inhibition of HIF-1α mRNA expression by 3-HT (Figure 4D). Following exposure to cobalt chloride (CoCl_2_) -induced hypoxia, the administration of 3-HT led to a dose-dependent decrease in the expression of HIF-1α protein. However, upon treatment with pyruvate, the impact of 3-HT on HIF-1α protein expression was not observed (Figure 4E). These results indicated that 3-HT inhibited the expression of HIF-1α by suppressing the activity of α-enolase.

### 2.5. 3-HT Negatively Regulates HIF-1α through the Activation of AMPK

To investigate the mechanism by which 3-HT regulates the expression of HIF-1α through the inhibition of α-enolase, an examination of AMPK activity was conducted to determine its role as a potential negative regulator of HIF-1α. Increased glycolysis resulted in a reduction in phospho-AMPK expression. When glucose was added, 3-HT rescued the AMPK activity in a dose-dependent manner. The supplementation of pyruvate did not lead to a significant effect of 3-HT on phospho-AMPK levels, as illustrated in Figure 5A. To demonstrate the alternation of AMPK activity, we evaluated the intracellular ATP levels. Consequently, treatment with glucose resulted in a reduction of the ATP levels induced by glycolysis in the presence of 3-HT. When pyruvate was exposed to treatment, the ATP levels were not affected by 3-HT (Figure 5B). The results suggest that 3-HT inhibited the activity of HIF-1α by stimulating AMPK, a mechanism controlled by glycolysis.

## 3. Discussion

*Salvia miltiorrhiza* is extensively used as an herbal remedy in traditional Chinese medicine. The plant belongs to the Lamiaceae family, and its medicinal components include the roots or rhizomes [18]. The chemical compounds extracted from *Salvia miltiorrhiza* can be classified into two main groups: (i) fat-soluble tanshinone compounds, predominantly Tanshinone I and Tanshinone IIA, and (ii) water-soluble phenolic compounds, primarily including salvianolate [19]. Diue to the growing clinical demand for tanshinones, there has been a surge in research aimed at enhancing the effectiveness of tanshinones and maximizing their pharmacological impacts through the exploration of structure-activity relationships [20]. Several studies have demonstrated that the chemical components of *Salvia miltiorrhiza*, particularly tanshinone IIA, exhibit potent antitumor characteristics such as inhibiting cell proliferation, arresting the cell cycle, triggering apoptosis, and promoting autophagic cell death in different tumor cell varieties [18]. Tanshinone IIA serves as a representative compound in *S. miltiorrhiza*. Nevertheless, because of its limited water solubility, the water-soluble derivative sodium tanshinone IIA sulfonate has been authorized by the Chinese National Medical Products Administration (NMPA) for the management of cardiovascular disease [21]. Tanshinones possess the ability to permeate the blood-brain barrier (BBB) due to their favorable lipophilic properties and low molecular weight. For example, research studies have indicated that tanshinone IIA [22], cryptotansinone [23,24], and tansinone I [25] demonstrate neuroprotective characteristics. Among these compounds, Tanshinone I selectively inhibits the expression of pro-inflammatory genes in activated microglia among these compounds. It also prevented nigrostriatal dopaminergic neurodegeneration in a mouse model of Parkinson’s disease [26]. However, there have been no studies conducted on the anticancer activity of 3-Hydroxytanshinone and its underlying mechanism of action.

Hypoxia, defined as a decrease in oxygen levels, is a common and significant feature observed in most solid tumors [27]. The hypoxic conditions play a crucial role in the expression of hypoxia-inducible factor (HIF). The transcriptional regulation of hundreds of genes is controlled by HIF-1α [28]. Cellular molecules linked to cancer development, such as vascular endothelial growth factor (VEGF) and glucose transporter 1 (GLUT-1), are regulated by HIF-1α activity [29]. Therefore, we investigated the most potent compound among tanshinone derivatives in terms of inhibiting the activity of the HIF-1α transcription factor. The study revealed that 3-HT isolated from *Salvia miltiorrhiza* regulates the transcriptional activity and the expression of hypoxia-inducible factor 1-alpha (HIF-1α) under hypoxic conditions. 3-HT also decreased the transcriptional levels of VEGF and GLUT-1. Consequently, we proposed that 3-HT regulates the expression of VEGF and GLUT-1 molecules, which are expressed by HIF-1α under hypoxic conditions.

Hypoxic conditions have been shown to stimulate angiogenesis within the tumor microenvironment. Activation of HIF-1α has been demonstrated to accelerate cancer advancement by facilitating angiogenesis [4]. The inhibition of angiogenesis in solid tumors has been associated with the absence of HIF-1α in the endothelium, as demonstrated in knockout mouse models [30]. The HIF-1 inhibitor effectively impeded angiogenesis at the cellular level, as demonstrated in a study [31]. Previous research has shown that tanshinone IIA exhibits anti-angiogenic properties in human umbilical vein endothelial cells and endothelial progenitor cells [32,33]. In this investigation, we present findings indicating that 3-HT reduced angiogenesis both in vitro and ex vivo. Furthermore, the expression level of VEGF was reduced by 3-HT. The findings of our study suggest that 3-HT effectively inhibits angiogenesis both in vitro and ex vivo by targeting HIF-1α, a key regulator in this process.

Particularly in glucose metabolism, tumor cells depend on unique metabolic pathways to support their rapid proliferation [34]. The Warburg effect is a phenomenon that enables cancer cells to utilize aerobic glycolysis for energy generation, even in the presence of adequate oxygen levels. This process results in an increased absorption of glucose and the consequent generation of lactate [35]. Under standard hypoxic conditions, the majority of eukaryotic cells exhibit the capacity to shift their primary metabolic strategy from primarily relying on mitochondrial respiration to upregulating glycolysis. This adaptation enables the cells to sustain ATP levels. The regulation of this metabolic switch can be influenced by various pathways, one of which involves the hypoxia-inducible factor-1α (HIF-1α), known for its role in promoting the upregulation of glycolytic enzymes [36]. In a previous study, tanshinone IIA inhibited glycolysis in non-small-cell lung cancer cells by modulating the transcription factor SIX1 [37]. The mRNA encoding α-enolase is subject to transcriptional regulation by MUC1, HIF-1α, and SIX1 [38]. In this study, the target protein for 3-HT was determined to be α-enolase by employing biotinylated 3-HT in combination with LC-MS/MS analysis. The results of our study also demonstrate that 3-HT inhibited the catalytic activity of α-enolase as observed in an enzymatic assay. In our experiments aimed at confirming the specificity of 3-HT’s binding to α-enolase, we have shown that purified α-enolase successfully bound to biotinylated 3-HT. However, following the co-administration of both 3-HT and biotinylated 3-HT, no detectable levels were observed. This discovery prompted us to formulate a hypothesis that suggests 3-HT exhibits a higher binding affinity compared to biotinylated 3-HT. Our findings indicate that the inhibition of enolase1 leads to the suppression of HIF-1α transcriptional processes.

According to a recent study, extracellular enolase1 has been found to regulate the expression of HIF-1α in a multiple myeloma cell line [39]. We conducted a glycolysis rescue experiment utilizing glucose and pyruvate, which serve as the substrate and end product of glycolysis, respectively. Our research was conducted to examine the correlation between α-enolase, a pivotal enzyme in glycolysis, and HIF-1α, a transcription factor responsible for regulating the cellular response to hypoxia. The study revealed that siRNA targeting α-enolase led to a reduction in the expression of HIF-1α in HeLa cells, irrespective of the hypoxic conditions. Moreover, upon exposure to glucose, the primary substrate for glycolysis, the administration of siRNA targeting α-enolase resulted in the suppression of the HIF-1α expression by impeding glycolytic processes. However, the administration of pyruvate, a byproduct of glycolysis, did not impact the expression of HIF-1α when siRNA targeting α-enolase was applied. This indicates that α-enolase modulates the expression of HIF-1α via glycolysis. Several studies have documented the regulation of α-enolase by HIF-1α [40,41,42]. Our findings revealed that the inhibition of α-enolase resulted in the downregulation of the HIF-1α expression. Moreover, it was observed that 3-HT inhibited the activity of α-enolase and modulated the expression of HIF-1α in HeLa cells by simulating the RNA interference of enolase1. From this perspective, our study offers insights into the mechanism through which 3-HT inhibits glycolysis by obstructing α-enolase, consequently regulating the expression of HIF-1α.

To elucidate the mechanism underlying the α-enolase-HIF-1α axis, our study focused on the cellular energy sensor AMPK. As a prominent glycolytic enzyme pivotal in ATP production, the malfunction of α-enolase is expected to exert a substantial influence on energy metabolism [43]. AMPK serves as a pivotal energy sensor and is essential for the regulation of metabolic processes in both normal and cancerous cells [44]. The activation of AMPK is recognized for its role in diminishing protein synthesis through the phosphorylation of TSC1/2, which leads to the inhibition of the TOR pathway, consequently suppressing the mTOR pathway [45]. In a previous study, it was found that the activation of AMP-activated protein kinase by AICAR, an AMPK activator, inhibits the expression of HIF-1α induced by insulin and IGF-1. The observations indicated that the expression of HIF-1α was regulated by insulin and IGF-1 through an mTOR/translation-dependent pathway [46]. Upon treatment with glucose or pyruvate, the activation of glycolysis inhibited phospho-AMPK. Our findings indicate that the inhibition of glycolysis through the blocking of α-enolase activity by 3-HT treatment resulted in the activation of AMPK. This current study presents, to the best of our knowledge, the first demonstration of α-enolase interference leading to HIF-1α inhibition via phospho-AMPK expression.

## 4. Materials and Methods

### 4.1. Cell Culture

HeLa cells were obtained from the American Type Culture Collection (ATCC, Manassas, VA, USA) and incubated in Dulbecco’s modified Eagle’s medium (DMEM; Thermo Fisher Scientific, Waltham, MA, USA) with 10% fetal bovine serum (FBS; Gibco) and antibiotics (100 U/mL of penicillin and 100 μg/mL streptomycin; Thermo Fisher Scientific Inc.). Human umbilical vein endothelial cells (HUVECs), purchased from ATCC, were cultured for 3–10 passages in human endothelial-SFM (Gibco) with 10% FBS. Hypoxia was induced using a hypoxia-mimetic reagent, CoCl_2_, for 12–18 h. The cells were grown under various conditions in a gas-controlled chamber (Thermo Fisher Scientific Inc., Waltham, MA, USA) with 1% O_2_, 94% N_2_, and 5% CO_2_ for hypoxic stimulation.

### 4.2. Reagents

CoCl_2_ was purchased from Sigma-Aldrich (St. Louis, MO, USA). To establish a stable cell line, antibiotics (hygromycin and puromycin) were procured from Invitrogen (Waltham, MA, USA). The primary antibody against HIF-1α was obtained from BD Biosciences (San Diego, CA, USA). Antibodies against p-AMPK, AMPK, and α-enolase were purchased from Cell Signaling Technology (Beverly, MA, USA). All other antibodies, including those against actin, were purchased from Santa Cruz Biotechnology (Dallas, TX, USA).

### 4.3. Chemicals

3-HT, biotinyl-3-HT, and tanshinone derivatives were generously provided by the Korea Research Institute of Chemical Technology (KRICT). 3-HT and biotinyl-3-HT were produced according to the methods outlined in a previous study [47].

### 4.4. Stable Cell Lines and Luciferase Assay

Two stable cell lines were used in the cell-based assays to identify inhibitors of HIF. RL vectors and HRE-FL vectors were obtained from the Promega Corporation (Madison, WI, USA). Using Lipofectamine 3000 (Invitrogen) and the HRE-FL plasmid to measure HIF activity and the RL plasmid to generate the internal control, stable cell lines were created [48]. Cells were transfected for 2 days and then incubated for 3 weeks in a DMEM medium containing hygromycin (150 μg/mL) and puromycin (5 μg/mL). A 3:1 mixture of the two stable cell lines (HRE-FL: RL) was used. A 96-well plate containing this mixed-cell line was seeded with 4000 cells per well and incubated overnight. Following the manufacturer’s instructions, the CoCl_2_-induced hypoxic cells were lysed, and the HIF activity was determined using a dual-luciferase assay kit (Promega). The luciferase activity was adjusted to be comparable to that of Renilla luciferase (RL). Luminescence was determined using a SpectraMax iD3 multi-mode microplate reader (Molecular Devices, Sunnyvale, CA, USA).

### 4.5. Cell Proliferation Assay

At a density of 4000 cells per well, HeLa cells were seeded in triplicate onto 96-well plates. After 3-HT treatment, the cells were cultured for 2 days under normoxic conditions, and cell proliferation was assessed using a Cell Counting Kit-8 (Dojindo Molecular Technologies, Rockville, MD, USA) following the manufacturer’s instructions.

### 4.6. Real-Time PCR

The primers (Table 1) were selected using the online Primer3 design software (v. 0.4.0) [49]. The total RNA was isolated briefly using TRIzol reagent. cDNA was prepared from 1 µg of total RNA. First-strand cDNA was synthesized using the RevertAid First Strand cDNA Synthesis Kit (Thermo Scientific, Waltham, MA, USA) following the manufacturer’s protocol. SYBR Green-based qPCR was run using the QuantStudio™ 5 real-time PCR System (Thermo Scientific, MA, USA) and PowerUp™ SYBR™ Green Master Mix (Thermo Scientific, MA, USA). All reactions were run in triplicate, and the data were analyzed using the 2^−ΔΔCT^ method. Histone H2A type 3 (HIST3H2A) was utilized as an internal control gene. Statistical significance was determined using Student’s *t*-tests with HIST3H2A-normalized 2^−ΔΔCT^ values; differences were considered significant at *p* < 0.05.

### 4.7. Western Blot Analysis

Whole-cell lysates were prepared using RIPA lysis buffer (Cell Signaling Technology) supplemented with a protease inhibitor cocktail (Roche Diagnostics, Mannheim, Germany). The protein concentration was measured using a detergent-compatible (DC) protein assay kit (Bio-Rad, Hercules, CA, USA). The proteins were subjected to SDS-PAGE and transferred to PVDF membranes (Merck Millipore, Darmstadt, Germany). In a cold room at 4 °C, the membranes were treated with primary antibodies. After TBST washing, the samples were then incubated with the suitable HRP-conjugated secondary antibody for 1 h at room temperature. Clarity Western ECL Substrate (Bio-Rad) was used to develop the PVDF membranes. The proteins were visualized using a ChemiDoc XRS+ system (Bio-Rad). Actin was used as the internal control.

### 4.8. In Vitro Capillary Tube Formation Assay

96-well plates were coated with 25 μL of Matrigel (BD Biosciences) and then left polymerize for 1 h at 37 °C. On the surface of the Matrigel, HUVECs (5 × 10^3^ cells/well) were seeded with SFM containing 1% FBS, 5 ng/mL basic fibroblast growth factor (bFGF; Thermo Fisher Scientific), and 2.5 ng/mL epidermal growth factor (EGF; Thermo Fisher Scientific). For 12 to 16 h at 37 °C, the cells were co-treated with various doses of 3-HT and 200 μM CoCl_2_. Morphological changes and tube developments were photographed using an inverted microscope (10×).

### 4.9. Chorioallantoic Membrane Assay

The chorioallantoic membrane (CAM) assay was performed as previously described [50]. For four days, fertilized chicken eggs were incubated at 37 °C in a humid incubator. A hypodermic needle was used to extract approximately 4–5 mL of egg albumin, facilitating the separation of the CAM and yolk sac from the shell membrane. Thermanox coverslips (Thermo Fisher Scientific) with compounds loaded on them were positioned on the CAM surfaces on day 5 after the shell membrane had been removed. After 2 days, Intralipose (Green Cross, Yongin, Republic of Korea) was injected beneath the CAM, and the membrane was then captured on camera. As a positive control, a potent anti-angiogenic substance known as retinoic acid (RA) was administered. The inhibition rate was expressed as a percentage by dividing the number of inhibited eggs by the total number of surviving eggs.

### 4.10. Target Identification

Before using Fast Protein Liquid Chromatography (FPLC), we installed the 1 mL streptavidin column (Amersham). Equilibration was achieved by passing 5 mL of a 1:1 mixture of DMSO and protein binding buffer through at a flow rate of 0.2 mL/min. After equilibration, 1 mL of 1 mg/mL biotinylated 3-HT was mixed with 1 mL of protein binding buffer (PBB; 25 mM HEPES, 1M NaCl, 10 mM imidazole, 10% glycerol, pH 7.5) and injected into a total volume of 2 mL. After binding the compound to the column, the column was washed with 30 mL of buffer, and then 2 mL of HeLa cell lysate protein was injected at a concentration of 15 mg/mL. After washing with 18 mL of buffer, 2 mL of 3-HT was injected at a concentration of 0.75 mg/mL to elute the protein. After collecting 0.5 mL of the protein fraction, the bound proteins were denatured and resolved using a 15% SDS-PAGE gel. The visualization of bound proteins was achieved using a silver staining kit (Elpis Biotech, Daejeon, Republic of Korea) following the manufacturer’s protocol. The band of interest was excised from the gel under sterile conditions, destained, and digested with trypsin. The amino acid sequence was determined using LC-MS/MS (nano-UPLC and LTQ Orbitrap XL), which was conducted by Proteome Tech (Seoul, Republic of Korea) as per our request.

### 4.11. Dotting Assay

A nitrocellulose membrane was dotted with purified α-enolase at each concentration and incubated at 4 °C for 15 min. After washing with TBST, the membrane was blocked in a 5% BSA solution for 2 h. Subsequently, the indicated compound or DMSO was applied to the membrane in the 5% BSA solution for 4 h at room temperature. After washing with TBST, the membrane was incubated with a secondary antibody (anti-biotin-HRP) in 5% BSA at room temperature for 2 h. Subsequently, the membrane was developed using Clarity Western ECL Substrate and visualized using the ChemiDoc XRS+ system.

### 4.12. Surface Plasmon Resonance Assay

Surface plasmon resonance (SPR) was applied as described [51] to investigate the binding of 3-HT to the potential target protein. Binding studies were conducted at 25 °C using a ProteOn XPR36 protein interaction array system (Bio-Rad). For measuring the kinetics of biotinyl-3-HT, an NLC chip was coated with biotinylated-3-HT. For immobilization, the proteins were diluted in a 10 mM acetate buffer (pH 5.0). The protein’s binding to the target compound was tested by flowing five concentrations ranging from 484 to 16 nM at a rate of 50 μL/min for 120 s over the compound-coated chip and monitoring dissociation for 600 s. The data were analyzed using ProteOn Manager 2.1.

### 4.13. α-Enolase Enzymatic Activity Assay

α-enolase enzymatic activity assay was performed using the Enolase Activity Assay Kit from Sigma-Aldrich. First, the reaction mixture was prepared according to the kit protocol. Recombinant enolase was plated in each well with DMSO or 3-HT, and 50 μL of the corresponding reaction mix was added to each well. The reaction was measured at 340 nm over 10 min at 30 s intervals using a multifunctional microplate reader (SpectraMax iD3).

### 4.14. ATP Assay

ATP levels were assessed using a luciferin/luciferase-based ATP assay kit (No. 11699709001, Roche Diagnostics). Briefly, HeLa cells were cultured at 3 × 10^4^ cells/mL in a 100 mm dish at 37 °C overnight. After changing the Krebs buffer, the samples were pre-treated with 3-HT for 1 h, then harvested and lysed following the assay kit protocol. The ATP standard was serially diluted in dilution buffer to a concentration ranging from 10^−6^ M and 10^−12^ M ATP. The samples, containing the same amount of cell lysis solution, were then incubated at 25 °C for 5 min. Appropriate amounts of the luciferase reagents were added to the samples, and measurements were taken every second from 1 s to 10 s using a multifunctional microplate reader (SpectraMax iD3).

### 4.15. Statistical Analysis

Quantitative values are presented as mean ± S.D. The results from one example experiment are displayed. The experiments were conducted three to five times. Using the Student’s *t*-test, we examined the significance of differences. *p* < 0.05 was considered to indicate statistical significance.

## 5. Conclusions

This study represents the inaugural investigation into the molecular mechanisms of 3-HT, to the best of our understanding. We investigated the effects of 3-HT on the inhibition of HIF-1α during hypoxia. Under hypoxic conditions, 3-HT suppressed the transcription of HIF-1α mRNA and the expression of its protein without inducing cytotoxic effects at the indicated concentrations. In in vitro and ex vivo studies, 3-HT demonstrated a reduction in angiogenesis by inhibiting VEGF. 3-HT was found to bind to α-enolase, a protein that plays a role in regulating the expression of HIF-1α under hypoxic conditions. The activation of AMPK is regulated by glycolysis, and 3-HT inhibits glycolysis by targeting α-enolase. This inhibition enhances AMPK activity, leading to a reduction in HIF-1α expression. These findings suggest that 3-HT has the potential to serve as a therapeutic agent for the purpose of targeting tumor angiogenesis and development (Figure 6).

## Figures and Tables

**Figure 1 molecules-29-02218-f001:**
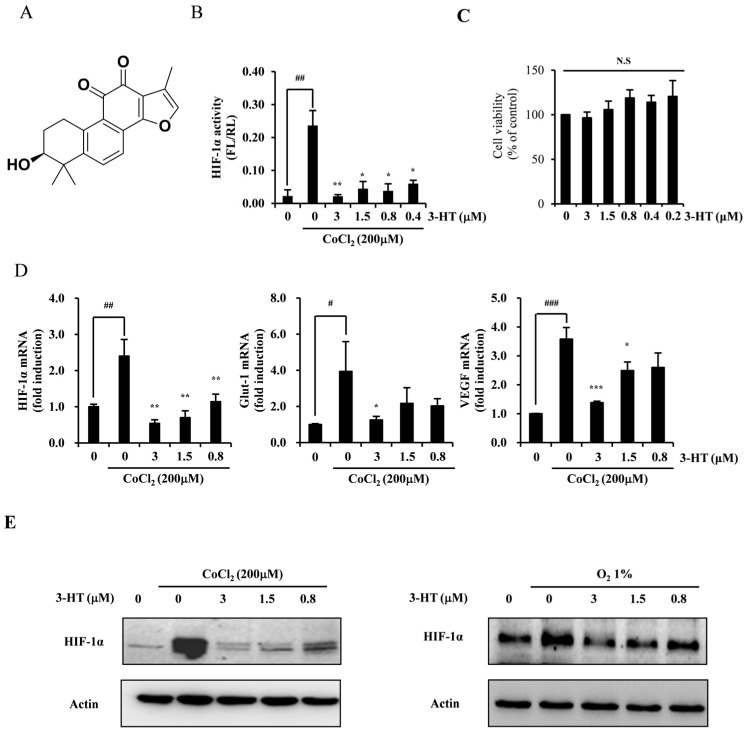
3-Hydroxytanshinone (3-HT) inhibits HIF-1α expression under hypoxia. (**A**) Molecular structure of 3-HT. (**B**) A mixture (3:1) of two stable cell lines (HeLa-hypoxia response element [HRE]-firefly luciferase [FL] and HeLa-CMV-Renilla luciferase [RL]) was plated into 96-well plates at a density of 4000 cells per well. The cells were incubated for 16 h in the presence of CoCl_2_ with the indicated concentrations of 3-HT. Luciferase activity was determined. Activity is expressed as fold induction relative to the activity in the absence of CoCl2. HeLa-CMV-RL activity was used to normalize luciferase activity. ^##^ *p* < 0.01 versus negative control; * *p* < 0.05, ** *p* < 0.01 versus positive control. (**C**) The effects of 3-HT on HeLa cell viability were evaluated using the CCK-8 assay. “N.S” indicates not significant (*p* > 0.05) (**D**) After pre-treatment with the indicated concentrations of 3-HT for 1 h, cells were treated with CoCl2 (200 μM) for 16 h. mRNA expression was then analyzed via real-time PCR. ^#^ *p* < 0.05, ^##^ *p* < 0.01, ^###^ *p* < 0.001 versus negative control; * *p* < 0.05, ** *p* < 0.01, *** *p* < 0.001 compared to the positive control. (**E**) The effects of 3-HT on HIF-1α expression were evaluated through Western blot analysis under conditions of CoCl_2_ treatment or 1% oxygen. Actin was used as the internal control.

**Figure 2 molecules-29-02218-f002:**
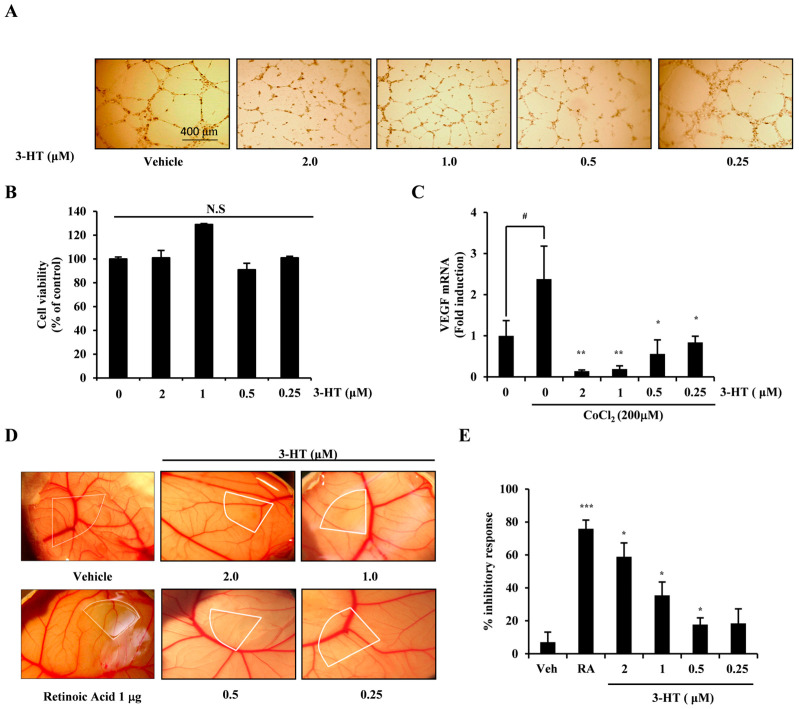
3-HT suppresses hypoxia-induced angiogenesis in both in vitro and ex vivo. (**A**) HUVECs were cultured at a density of 5 × 10^3^ cells per well on matrigel in human endothelial-serum free medium (SFM) with CoCl_2_ and the indicated concentration of 3-HT for 16 h. Subsequently, the cells were fixed and examined under an inverted microscope (400 μm scale bar). (**B**) The effects of 3-HT on HUVECs viability were assessed through the CCK-8 assay. (**C**) HUVECs were stimulated for 16 h with CoCl_2_ in the presence of the appropriate concentration of 3-HT. Real-time PCR was used to measure mRNA expression after extracting the total RNA using the TRIzol reagent. ^#^ *p* < 0.05 relative to the negative control; * *p* < 0.05, ** *p* < 0.01 relative to the positive control. (**D**) The chorioallantoic membrane (CAM) of fertilized chicken eggs was placed on uniformly sized sterilized coverslips loaded with the indicated concentration of 3-HT. The coverslips were left in place for 2 days. A digital camera was used to capture the development of neovessels. White areas on the CAM surfaces indicate where 3-HT has been applied. (**E**) Activity was quantified based on the proportion of positive eggs relative to the total number of eggs tested. * *p* < 0.05, *** *p* < 0.001 versus vehicle.

**Figure 3 molecules-29-02218-f003:**
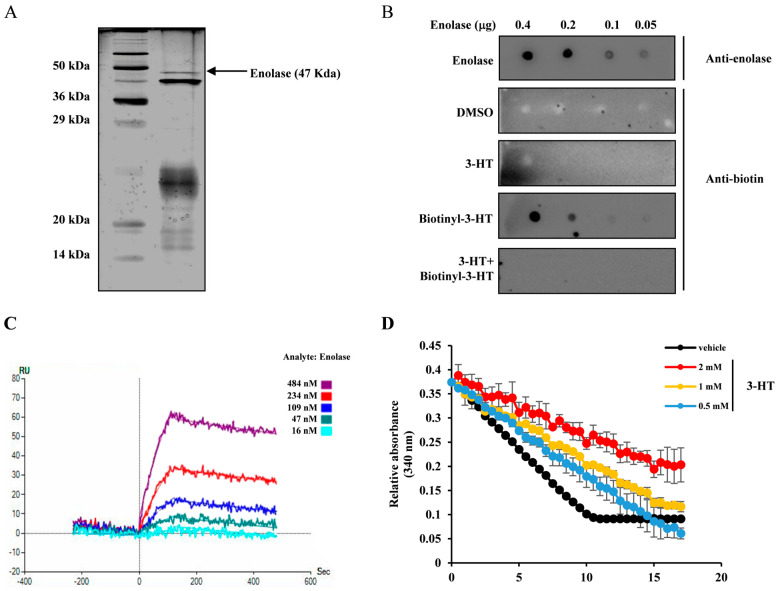
3-HT binds to the α-enolase protein, resulting in the inhibition of its enzymatic activity. (**A**) Silver staining of biotinyl-3-HT. For fast protein liquid chromatography, biotinyl-3-HT was injected into a 1 mL streptavidin column. After washing, 30 mg of protein was injected. After further washing, each 0.5 mL fraction was eluted. SDS-PAGE was performed using a 15% gel with 25 μL loading, followed by silver staining and LC-MS/MS. (**B**) Dotting assay of 3-HT. Purified enolase protein was prepared. Each concentration of enolase was spotted onto a nitrocellulose membrane, which was then incubated at 4 °C for 15 min. After washing twice with TBST, a 5% bovine serum albumin (BSA) in TBST was used as a blocking buffer for 2 h. The enolase dots were then treated with either anti-biotin compounds or anti-enolase antibodies in 5% BSA for 4 h. Following three washes, anti-biotin-HRP or anti-goat-HRP was applied as a secondary antibody with 5% BSA for 2 h. (**C**) Surface plasmon resonance (SPR) analysis of 3-HT binding to α-enolase. Biotinyl-3-HT was immobilized on a chip, and enolase was injected into the flow cells. (**D**) Enolase enzymatic assay for 3-HT. Purified α-enolase was mixed with DMSO or 3-HT. The reaction was detected at 340 nm over 10 min at 30-s intervals using a multifunctional microplate reader.

**Figure 4 molecules-29-02218-f004:**
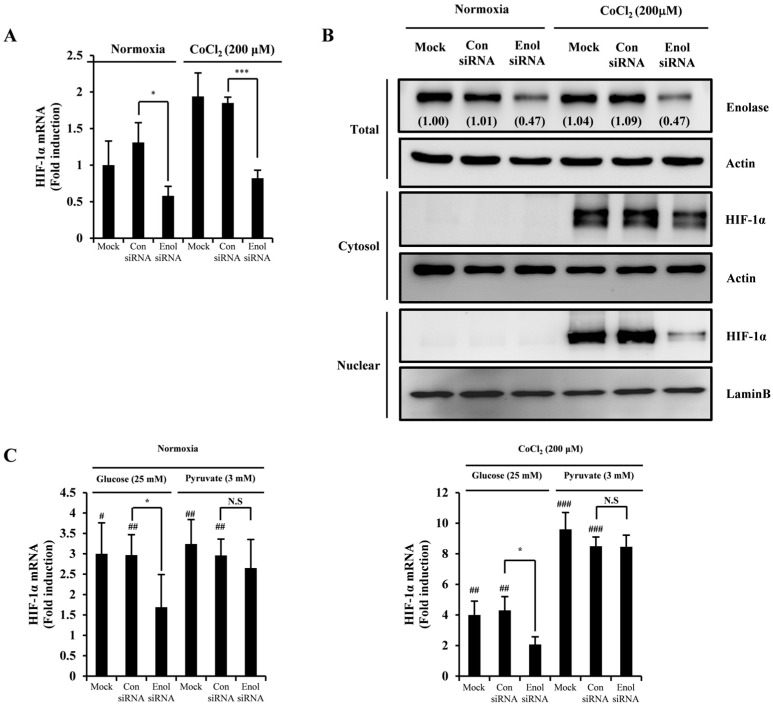
3-HT inhibits the expression of HIF-1α by directly regulating the activity of α-enolase. (**A**) After transfection with siRNA against α-enolase for 2 days, HeLa cells were stimulated with CoCl_2_ (200 μM) for 6 h. mRNA expression was measured using real-time PCR. * *p* < 0.05, *** *p* < 0.001 versus the control siRNA group. (**B**) After transfection with siRNA against α-enolase for 2 days, the cells were treated with CoCl_2_ for 12 h and then lysed using NE-PER Nuclear and Cytoplasmic Extraction reagents. Protein samples were analyzed via Western blotting. Actin was used as the internal control. (**C**) Cells were transfected with siRNA against α-enolase for 2 days and treated with glucose and pyruvate in the presence of CoCl_2_ (200 μM) for 6 h. The total RNA was harvested using TRIzol reagent. The mRNA expression was evaluated using real-time PCR. ^#^ *p* < 0.05, ^##^ *p* < 0.01, ^###^ *p* < 0.001 versus the negative control; * *p* < 0.05 versus the control siRNA group. (**D**) The effect of 3-HT on HIF-1α mRNA expression was measured using real-time PCR under conditions of CoCl_2_ (200 μM) or 1% oxygen for 6 h in the presence of either glucose or pyruvate. ^##^ *p* < 0.01, ^###^ *p* < 0.001 versus the negative control; * *p* < 0.05, ** *p* < 0.01, *** *p* < 0.001 versus the positive control. (**E**) The effects of 3-HT on HIF-1α protein expression were evaluated using Western blotting with glucose or pyruvate in the presence of CoCl_2_ (200 μM). Actin was used as the internal control.

**Figure 5 molecules-29-02218-f005:**
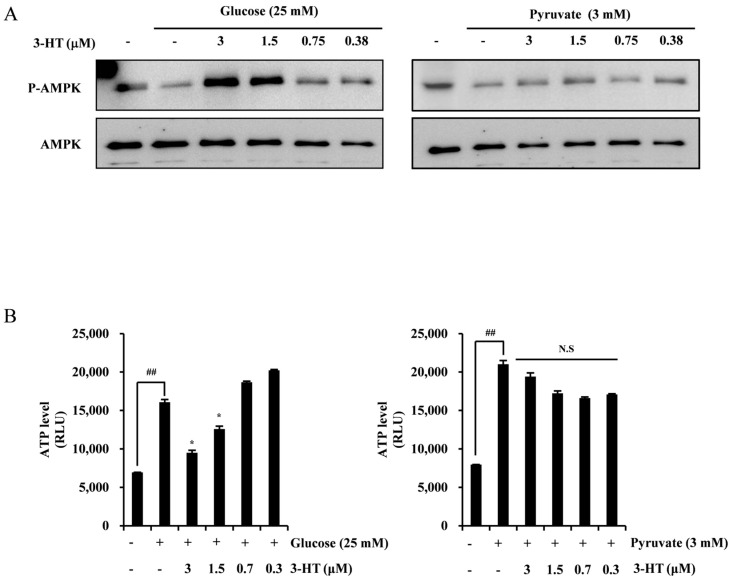
3-HT negatively regulates HIF-1α by activating AMPK. (**A**) HeLa cells were pre-treated with the recommended concentration of 3-HT for 1 h, followed by 4 h of treatment with either glucose or pyruvate. The protein levels were analyzed via Western blotting. AMPK was used as the internal control. (**B**) The effects of 3-HT on HeLa cell ATP levels were measured using an ATP assay kit with co-treatment of glucose or pyruvate. ^##^ *p* < 0.01 compared to the negative control; * *p* < 0.05 compared to the positive control.

**Figure 6 molecules-29-02218-f006:**
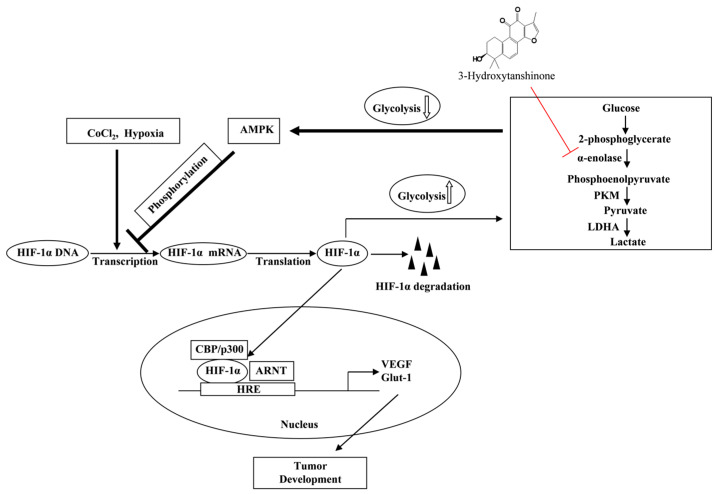
A proposed mechanism detailing the inhibition of α-enolase, which is regulated by AMPK phosphorylation, leading to the attenuation of HIF-1α in HeLa cells by 3-HT.

**Table 1 molecules-29-02218-t001:** Primer sequences used in this study.

Target Gene	Forward Primer (5′–3′)	Reverse Primer (5′–3′)
*Glut-1*	TGGATGTCCTATCTGAGCATCG	CTCCTCGGGTGTCTTGTCAC
*VEGF*	AACTTTCTGCTGTCTTGG	TTTGGTCTGCATTCACAT
*HIF-1α*	ACTTAAGAAGGAACCTGATG	TGGAGACATTGCCAAATTTA
*HIST3H2A*	CTTGACTCGGAAATGTCCGGTCG	AGTCAAGTACTCGAGCACCGCG

## Data Availability

The data presented in this study are available on request from the corresponding author.

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
