# Peer review of "3-Hydroxytanshinone Inhibits the Activity of Hypoxia-Inducible Factor 1-α by Interfering with the Function of α-Enolase in the Glycolytic Pathway"

_molecules, 2024, doi:10.3390/molecules29102218_

Round 1

Reviewer 1 Report

Comments and Suggestions for Authors

Abstract:

The authors used many undefined abbreviations. Also, you should define for the reader that tanshinones are traditional Chinese medicine with multiple vaiet of therapeutic effects.

In the section of discussion, the authors did not discuss the issue of bioavailability and hydrophobicity of tanshinones and possible chemical structure  modification to be tried at preclinical and clinical levels . An issue of great importance.

What is their cytotoxicity on normal cells?

.

Comments on the Quality of English Language

The English language is average

Author Response

We want to thank you and the reviewers for giving us the opportunity to revise our manuscript entitled “3-Hydroxytanshinone Inhibits the Activity of Hypoxia-Inducible Factor 1-α by Interfering with the Function of α-enolase in the Glycolytic Pathway.” (molecules-2974696) We appreciated the thoughtful comments. Please find our responses to the reviewers’ comments below.

â–ºReviewer 1’s comments:

Abstract:

The authors used many undefined abbreviations. Also, you should define for the reader that tanshinones are traditional Chinese medicine with multiple vaiety of therapeutic effects.

A: As recommended, we have added a list of abbreviations in the part of manuscript and tanshinone has been explicitly defined as a traditional Chinese medicine in the abstract section.

In the section of discussion, the authors did not discuss the issue of bioavailability and hydrophobicity of tanshinones and possible chemical structure modification to be tried at preclinical and clinical levels. An issue of great importance.

A: Thank you for this important suggestion. We have added the reviewer's comments to the discussion section.

What is their cytotoxicity on normal cells?

A: Thank you for your thorough review. A test was performed to assess the cytotoxic effects of tanshinone using Human Umbilical Vein Endothelial Cells (HUVECs). No toxicity was detected at the concentrations utilized in the study. Moreover, it was found that tanshinone demonstrated no toxicity in human bone marrow-derived endothelial progenitor cells (EPCs) at concentrations up to 30 μM, as reported in the articles. Based on these findings, it can be inferred that tanshinone does not exhibit toxicity towards normal cells.

Reference

  • Xing, Y., Tu, J., Zheng, L., Guo, L., & Xi, T. (2015). Anti-angiogenic effect of tanshinone IIA involves inhibition of the VEGF/VEGFR2 pathway in vascular endothelial cells. Oncology Reports, 33, 163-170. https://doi.org/10.3892/or.2014.3592
  • Tsai, M. Y., Yang, R. C., Wu, H. T., Pang, J. H., & Huang, S. T. (2011). Anti-angiogenic effect of Tanshinone IIA involves inhibition of matrix invasion and modification of MMP-2/TIMP-2 secretion in vascular endothelial cells. Cancer letters, 310(2), 198–206. https://doi.org/10.1016/j.canlet.2011.06.031
  • Lee, H. P., Liu, Y. C., Chen, P. C., Tai, H. C., Li, T. M., Fong, Y. C., Chang, C. S., Wu, M. H., Chiu, L. P., Wang, C. J., Chen, Y. H., Wu, Y. J., Tang, C. H., & Wang, S. W. (2017). Tanshinone IIA inhibits angiogenesis in human endothelial progenitor cells in vitro and in vivo. Oncotarget, 8(65), 109217–109227. https://doi.org/10.18632/oncotarget.22649

We heartily thank you and the reviewers again and hope that the revised version of our manuscript is now acceptable for publication in Molecules. We are looking forward to your positive response. Thank you.

sincerely yours,

Sik-Won Choi, Ph.D.

Reviewer 2 Report

Comments and Suggestions for Authors

This is an extremely interesting paper, very well researched with appropriate controls.  I have no doubt this will be an extremely well cited paper. Congratulations all round.  

Author Response

We want to thank you and the reviewers for giving us the opportunity to revise our manuscript entitled “3-Hydroxytanshinone Inhibits the Activity of Hypoxia-Inducible Factor 1-α by Interfering with the Function of α-enolase in the Glycolytic Pathway.” (molecules-2974696) We appreciated the thoughtful comments. Please find our responses to the reviewers’ comments below.

â–ºReviewer 2’s comments:

This is an extremely interesting paper, very well researched with appropriate controls.  I have no doubt this will be an extremely well cited paper. Congratulations all round.

A: We express our gratitude for the favorable feedback received regarding our research.

We heartily thank you and the reviewers again and hope that the revised version of our manuscript is now acceptable for publication in Molecules. We are looking forward to your positive response. Thank you.

sincerely yours,

Sik-Won Choi, Ph.D.

Reviewer 3 Report

Comments and Suggestions for Authors
  1. Regarding the potential anti-tumor effects of 3-Hydroxytanshinone both in vitro and in vivo, as well as its ability to inhibit angiogenesis, the authors should provide supporting data to substantiate these claims in the manuscript.

  2. For Figure 2, it is recommended that the authors include scale bars in the representative images

    so that the reader understands the actual dimensions depicted.
  3. In Figure 4B, the evidence supporting the successful targeting of α-enolase by siRNA appears to be weak. The α-enolase bands in the Western blot are overexposed, not within the linear grayscale range,

    and statistical analysis of α-enolase inhibition is lacking.
  4. In Figure 4B, if the authors aim to demonstrate the expression levels of HIF1a in the cytoplasm and nucleus respectively, it is advisable to include corresponding loading controls such as β-actin and Lamin B1 for the cytosol and nuclear fractions, respectively, to confirm the successful separation of these two components.
  5. The authors should ensure that all representative Western blot results are labeled with molecular weight markers, and they should pay attention to the quality of individual Western blot images, such as in Figure 5A.

Author Response

We want to thank you and the reviewers for giving us the opportunity to revise our manuscript entitled “3-Hydroxytanshinone Inhibits the Activity of Hypoxia-Inducible Factor 1-α by Interfering with the Function of α-enolase in the Glycolytic Pathway.” (molecules-2974696) We appreciated the thoughtful comments. Please find our responses to the reviewers’ comments below.

â–ºReviewer 3’s comments:

  1. Regarding the potential anti-tumor effects of 3-Hydroxytanshinone both in vitro and in vivo, as well as its ability to inhibit angiogenesis, the authors should provide supporting data to substantiate these claims in the manuscript.

A: We respect the opinion of the reviewer. We have elucidated the anti-angiogenic effect of 3-hydroxytanshinone in Figure 2 through in-vitro experiments using the HUVECs tube formation assay and ex-vivo experiments employing the CAM assay. We provide an overview of the anti-tumor effects linked to the inhibition of Hypoxia-Inducible Factor (HIF)-1α and elucidate the mechanisms that underlie the inhibition of HIF in these study.

  1. For Figure 2, it is recommended that the authors include scale bars in the representative images so that the reader understands the actual dimensions depicted.

A: Thank you for your insightful suggestion. As suggested, scale bars have been incorporated into the representative microscope image in Figure 2a. In the case of Figure 2d, the image was captured using a digital camera, which precluded the direct measurement of its physical dimensions.

  1. In Figure 4B, the evidence supporting the successful targeting of α-enolase by siRNA appears to be weak. The α-enolase bands in the Western blot are overexposed, not within the linear grayscale range, and statistical analysis of α-enolase inhibition is lacking.

A: Thank you for your thorough review. In accordance with your recommendation, we have substituted the western blot image with a brief exposure to demonstrate the knockdown effect induced by siRNA. We quantified the expression ratio of α-enolase by normalizing it to β-actin using data obtained from a western blot image.

  1. In Figure 4B, if the authors aim to demonstrate the expression levels of HIF1a in the cytoplasm and nucleus respectively, it is advisable to include corresponding loading controls such as β-actin and Lamin B1 for the cytosol and nuclear fractions, respectively, to confirm the successful separation of these two components.

A: Thank you for this suggestion. As recommended, we included the loading controls such as β-actin and Lamin B1 for the cytosol and nucleus fractions to confirm the successful separation of these compartments.

  1. The authors should ensure that all representative Western blot results are labeled with molecular weight markers, and they should pay attention to the quality of individual Western blot images, such as in Figure 5A.

A: Thank you for this important suggestion. During the immunoblot procedure, a prestaining ladder (Precision Plus Protein Standards Dual Color, Cat. No. #161-0374, Bio-Rad) is employed to avoid contamination of the marker in the western blot image. Upon transfer to the PVDF membrane, the marker becomes visible, facilitating the identification of the correct size by measuring it with a ruler to ensure the accuracy of the antibody size. As you mentioned, the molecular weight size marker was labeled and arranged on all Western blot images, which are provided below.

We heartily thank you and the reviewers again and hope that the revised version of our manuscript is now acceptable for publication in Molecules. We are looking forward to your positive response. Thank you.

sincerely yours,

Sik-Won Choi, Ph.D.

Round 2

Reviewer 3 Report

Comments and Suggestions for Authors

I have no more questions.